# Action mechanism of a novel agrichemical quinofumelin against *Fusarium graminearum*

**Qian Xiu[1†], Xiaoru Yin[1†], Yuanyuan Chen[2†], Ziyang Zhang[1], Yushuai Mao[1], Tianshi Wang[2], Jie Zhang[1], Mingguo Zhou[1]\*, Yabing Duan[1]\***

[1]State Key Laboratory of Agricultural and Forestry Biosecurity, College of Plant Protection, Nanjing Agricultural University, Nanjing, China; [2]College of Science, Nanjing Agricultural University, Nanjing, China

---

## eLife Assessment

In this **valuable** study, the authors show the physiological response and molecular pathway mediating the effect of quinofumelin, a developed fungicide with an unknown mechanism. The authors present **convincing** data suggesting the involvement of the uridine/uracil biosynthesis pathway, by combining in vivo microbiology characterization as well as in vitro biochemical binding results.

---

**\*For correspondence:**
mgzhou@njau.edu.cn (MZ);
dyb@njau.edu.cn (YD)

[†]These authors contributed equally to this work

**Abstract** Modern fungicides have made significant contributions to crop disease management, but the development of resistant fungal strains has caused their failure in disease control. Therefore, developing fungicides with novel action mechanisms is the most effective measure to manage resistance. Quinofumelin, a novel quinoline fungicide, exhibits exceptional antifungal activity against phytopathogens. However, there is currently no available information on its mechanism of action. Here, we used transcriptome and metabolome analysis to observe a co-enrichment pattern of differentially expressed genes (DEGs) and differentially accumulated metabolites (DAMs) within pyrimidine biosynthesis pathway (PBP), identifying down-regulation of dihydroorotate dehydrogenase (DHODH). Exogenous uridine monophosphate (UMP), uridine, or uracil (metabolites in PBP) successfully restored quinofumelin-induced inhibition of mycelial growth in *Fusarium graminearum* and *Fusarium asiaticum*. Additionally, the deletion of *FgDHODHII* was determined to be lethal; however, mycelial growth of ΔFgDHODHII mutants could be restored by adding UMP, uridine, or uracil. These findings indicate that the deficiencies in *FgDHODHII* are functionally equivalent to complete inhibition of its activity by quinofumelin. Finally, molecular docking, surface plasmon resonance (SPR), and microscale thermophoresis (MST) results strongly support the precise interaction between quinofumelin and FgDHODHII. Collectively, these findings provide compelling evidence for the involvement of de novo uracil biosynthesis as a mechanism of action for quinofumelin while identifying FgDHODHII as its specific target.

## Introduction

Agrochemical fungicides have a history spanning over 3000 years in the control of plant diseases, and have undergone a development process from traditional protective fungicides with multiple action sites to modern selective fungicides with single action site (*Lamberth, 2022*). The introduction of the first fully systemic seed treatment agent (carboxin, a succinate dehydrogenase inhibitor) in 1966 and the launch of the first broad-spectrum foliar systemic fungicide (benomyl, a tubulin polymerization inhibitor) in 1968 represent significant milestones in the advancement of modern selective fungicides

with single action site (*Lamberth et al., 2021*; *Morton and Staub, 2008*). Owing to their exceptional selectivity, potent activity, specific target site, and low toxicity, modern selective fungicides have gained widespread acceptance and witnessed an increasing market share year after year. However, like every coin has two sides, these types of fungicides encounter substantial challenges related to serious resistance issues when deployed in agricultural fields (*Lucas et al., 2015*). Numerous reports have been published on plant pathogens developing resistance to modern selective fungicides, which often result in failures or reductions in the effectiveness of chemical control against plant diseases and lead to decreased crop yields and quality (https://www.frac.info; *Duan et al., 2019*; *King et al., 2021*; *Mao et al., 2024*). Therefore, it is crucial to expedite the development of new-generation selective fungicides with innovative modes of action. Based on a single fungicide target, numerous modern selective fungicides can be developed (*Umetsu and Shirai, 2020*). For instance, 24 commercialized available fungicides have been created so far based on the respiratory chain complex II as the target. Similarly, another 20 commercialized fungicides have been developed targeting the respiratory chain complex III. Additionally, there are already 37 commercialized fungicides that specifically target the fungicide target C14-demethylase in ergosterol biosynthesis (https://www.frac.info). However, identifying specific targets that efficiently control plant pathogens while ensuring safety for non-target organisms remains a significant challenge. The rate of discovery for new targets lags far behind current demands.

According to the latest information released by the Fungicide Resistance Action Committee (FRAC), over 30 fungicide selective targets have been identified and acknowledged in the field of fungicide biological research spanning more than 130 years. Currently, these targets have been identified based on the successful development of fungicides and are all related proteins that are essential for pathogen growth or pathogenicity. Extensive identification and functional analysis of essential genes for growth or pathogenicity in plant pathogens have been achieved with rapid advancements in molecular biology techniques; however, there is no precedent for successfully developing commercialized fungicides targeting genes involved in growth or pathogenicity. The creation of highly active modern selective fungicides currently relies on the structural characteristics of existing targets and is accomplished through technologies such as AI drug discovery and design (AIDD) and computer-aided drug design (CADD) (*Yang et al., 2019*). The biological functions of drug targets are highly conserved, although variations may occur at critical amino acid sites during genetic evolution. Therefore, it is crucial to explore drug targets and analyze their structural characteristics in order to develop novel fungicides with enhanced activity and selectivity. Recently, the Fungicide Biology group from Nanjing Agricultural University proposed a new concept for targeting pesticides based on the principles of targeted drugs in the medicine field (*Zhong et al., 2021*). The objective is to create specialized targeted pesticides that exhibit higher activity, greater selectivity, stronger specialization, lower toxicity, and environmental friendliness by leveraging the structural characteristics of pesticide targets. This approach aims to provide essential technical support for scientifically controlling crop diseases.

Quinofumelin (CAS 861647-84-9) is a novel quinoline fungicide developed by Mitsui Chemicals Co., Ltd. Japan (*Ito et al., 2023*). Due to their significant bioactivity, quinoline compounds and their derivatives have found extensive applications in medicine and pesticides (*Tao et al., 2021*; *Wheeler et al., 2003*). In our previous study, quinofumelin not only exhibited excellent antifungal activity against the mycelial growth and spore germination of *F. graminearum*, but also inhibited the biosynthesis of deoxynivalenol (DON) (*Xiu et al., 2021*). In previous studies, quinofumelin was identified as a specific inhibitor of dihydroorotate dehydrogenase class II (DHODHII) in *Pyricularia oryzae*, suppressing fungal growth by blocking the conversion of dihydroorotate to orotate (*Higashimura et al., 2022*). Gene disruption of *DHODHII* (*PoPYR4*) further confirmed its critical role in fungal pathogenicity and validated it as the action target site of quinofumelin. However, previous investigations did not report any detailed methodologies for target identification. Additionally, there was a lack of information regarding the precise interaction between quinofumelin and DHODHII. In this study, we used transcriptomic and metabolomic analyses to investigate the co-enrichment patterns of DEGs and (DAMs). This approach enabled the identification of potential metabolic pathways were identified. Ultimately, the pyrimidine biosynthesis pathway was identified as the target. Therefore, this study unveils that the mechanism of action of quinofumelin involves de novo synthesis of uracil, thereby identifying DHODH as its specific target. As a selective fungicidal target, DHODH offers a valuable resource for the design and development of novel fungicides.

## Results

### Transcriptomic analysis of *F. graminearum* as affected by quinofumelin

RNA-seq experiments have emerged as the standard approach for quantifying and comparing gene expression levels across a wide range of species and conditions, providing critical insights into cellular phenomena (*Pertea et al., 2016*). To investigate the impact of quinofumelin on gene expression levels in *F. graminearum*, we performed an RNA-seq experiment. For quality assessment of the RNA-seq data, we conducted a comprehensive quality control analysis. In the principal component analysis, PC1 and PC2 accounted for 45.59% and 16.98% of the total variance, respectively. The distinct separation observed between the groups indicates significant differences in gene expression between the controls and experimental samples, which could be used for subsequent investigations (*Figure 1—figure supplement 1*). The clustering heat map effectively categorized the controls and experimental samples into distinct clusters based on their gene expression profiles, demonstrating robust biological replication with minimal intra-group variability and ensuring the reliability of the data. Moreover, notable inter-group dissimilarities highlighted variations among different treatment samples (*Figure 1C*).

The differential analysis unveiled a total of 234 significantly DEGs between the control and experimental samples. Among these findings, there were identified as up-regulated for a count of 97 genes while down-regulation was observed in case of another set comprising around 137 genes (*Figure 1A*). A graphical representation was employed to depict the statistical distribution pattern exhibited by these DEGs (*Figure 1B*). To gain deeper insights into gene functionalities, Gene Ontology (GO) and Kyoto Encyclopedia of Genes and Genomes (KEGG) enrichment analyses were performed on these DEGs. The results showed that they are notably enriched in seventeen distinct GO terms spanning across two major categories: eleven molecular functions along with six processes (*Figure 1D*). Furthermore, the enrichment analysis highlighted significant associations with various functional attributes, including transition metal ion binding, oxidoreductase activity, modified amino acid binding, amide binding, monooxygenase activity, and heme binding. Moreover, the enriched biological processes are primarily involved in transcriptional regulation driven by nucleic-enabled mechanisms, as well as monocarboxylic acid metabolism and RNA biosynthesis. Additionally, the KEGG enrichment analysis identified specific pathways associated with thiamine metabolism, tryptophan metabolism, nitrogen metabolism, amino acid sugar and nucleotide sugar metabolism, pantothenic acid and CoA biosynthesis, as well as nucleotide sugar production compound synthesis (*Figure 1E*). In the GO and KEGG enrichment analyses, the detailed counts of down- and up-regulated DEGs for each metabolic pathway are provided in *Supplementary file 1*.

### Metabolomic analysis of *F. graminearum* as affected by quinofumelin

According to the findings from the KEGG enrichment analysis of DEGs derived from transcriptomic data, these DEGs are associated with metabolism pathways as above described. To specifically investigate the metabolic pathways involved in the action mechanism of quinofumelin, we performed further metabolomic experiments. In metabolomics principal component analysis, the contribution rate of PC1 was determined to be 76.6% in positive ion mode and 51.2% in negative ion mode, respectively. Moreover, the quality control (QC) samples exhibited highly concentrated results, indicating excellent instrument stability and remarkable data repeatability. Furthermore, a distinct separation between groups was observed, particularly prominent in the positive ion mode, indicating significant variations in metabolites among different groups that warrant further investigation (*Figure 1—figure supplement 2A, B*). The findings from Pearson correlation analysis were consistent with those obtained from principal component analysis (*Figure 1—figure supplement 2C, D*). Strong correlations were identified within samples belonging to the same group, while weak correlations were observed between different groups.

Metabolomics analysis revealed a total of 713 differential metabolites, with 14 up-regulated and 699 down-regulated in the positive ion mode, as well as 230 differential metabolites, including 103 up-regulated and 127 down-regulated in the negative ion mode (*Figure 1—figure supplement 2A*; *Figure 2—figure supplement 1A*). Based on the classification of the identified 943 different metabolites, it was observed that amino acids and their derivatives accounted for the highest proportion (17%), followed by organic acids and their derivatives (11%), benzene and substituted derivatives (11%), aldehydes, ketones, esters (11%), and heterocyclic compounds

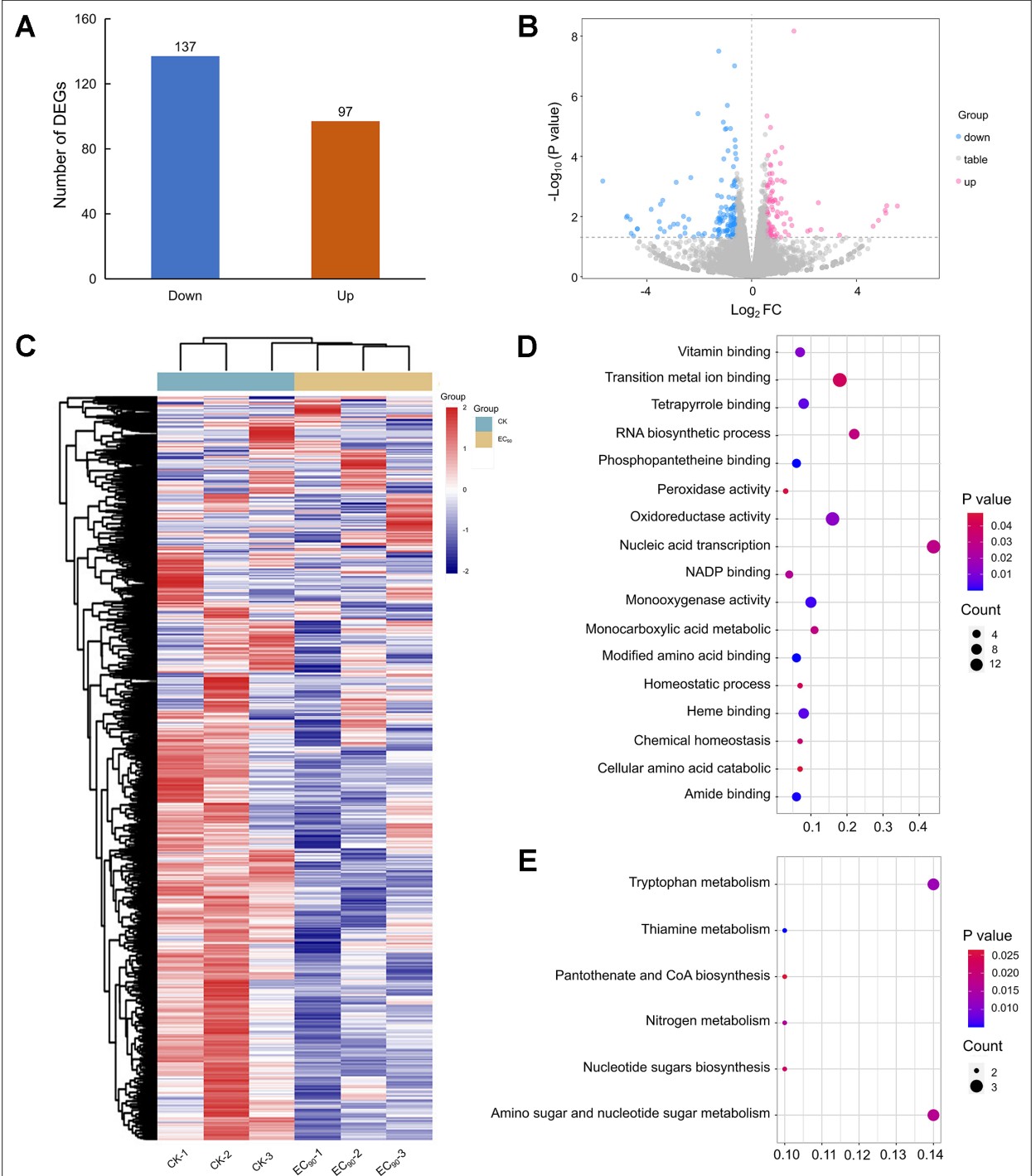

**Figure 1.** Analysis of transcriptome sequencing data in *F.graminearum* PH-1 treated with quinofumelin. (**A**) Bar chart illustrating the comparison of differentially expressed genes between $EC_{90}$ treatments and the control. (**B**) Volcano plot depicting up-regulated differentially expressed genes (DEGs) as red scattered dots and down-regulated DEGs as blue scattered dots. (**C**) Cluster heat maps displaying gene relative expression values using RNA-seq data scale-standardized values. (**D**) Bubble map presenting the results of Gene Ontology (GO) enrichment analysis for DEGs. (**E**) Bubble map showcasing the findings of Kyoto Encyclopedia of Genes and Genomes (KEGG) enrichment analysis for DEGs.

The online version of this article includes the following figure supplement(s) for figure 1:

**Figure supplement 1.** Principal component analysis plots between the experimental (EC90: 1 μg/mL quinofumelin) and control (CK) samples.

**Figure supplement 2.** Cluster analysis of the metabolite group.

(10%) (*Figure 2c*). The volcano plot visually represents the distribution of statistical differences in metabolite abundance between the two groups (*Figure 1—figure supplement 2B*; *Figure 2—figure supplement 1B*).

KEGG pathways enriched by differential metabolites primarily include arachidonic acid metabolism, pyrimidine metabolism, alanine, aspartate, and glutamate metabolism, purine metabolism, citrate cycle, arginine and proline metabolism, fructose and mannose metabolism, nitrogen metabolism, and metabolic pathways (*Figure 2D*). The classification of enrichment pathways for differential metabolites revealed that the majority of these metabolites were associated with nucleotide metabolism (25%), amino acid metabolism (25%), carbohydrate metabolism (25%), and energy metabolism (12.5%) (*Figure 2—figure supplement 1C*).

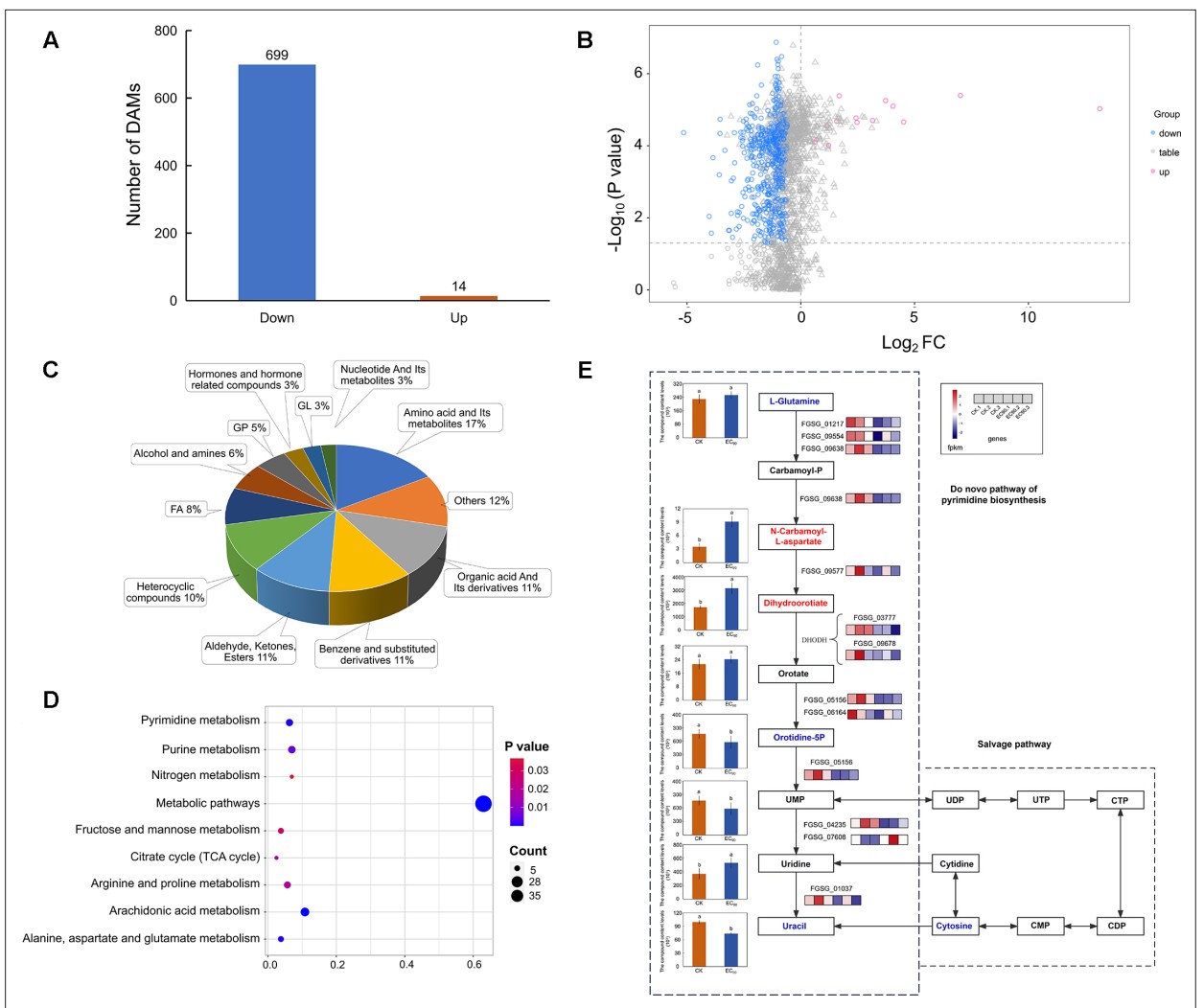

**Figure 2.** Difference analysis and enrichment analysis of metabolome data in *F. graminearum* PH-1 treated with quinofumelin. (**A**) Bar chart illustrating the number of differential accumulation metabolites (DAMs) quantity in experimental groups (EC$_{90}$) and control groups (CK) in positive ion mode. (**B**) Volcanic map of all metabolites in positive ion mode. The red scatter points represent up-regulated DAMs, while the blue scatter points represent down-regulated DAMs. (**C**) Classification and proportion of DAMs. (**D**) Bubble diagram of differential accumulation metabolite Kyoto Encyclopedia of Genes and Genomes (KEGG) enrichment pathway. (**E**) Pyrimidine de novo biosynthesis pathway in *Fusarium graminearum*. The left box plots represent the relative content of eight metabolite samples, while the right box represents the relative content of six gene samples. The color gradient from navy blue to firebrick red indicates a progression from low to high content.

The online version of this article includes the following figure supplement(s) for figure 2:

**Figure supplement 1.** Difference analysis of metabolic group data.

**Figure supplement 2.** DHODHII gene expression in *F. graminearum* is affected by quinofumelin.

## Conjoint analysis of transcriptome and metabolome integration

Through transcriptome and metabolome data analysis, we observed a co-enrichment of differentially expressed genes and differentially accumulated metabolites in the pyrimidine metabolism pathway. In the expression calorigrams of metabolites and genes depicting the de novo biosynthesis pathway of pyrimidine, we identified down-regulation of dihydroorotate dehydrogenase (DHODH) in the experimental groups. Through in vitro RT-qPCR verification, it was found that after treating with quinofumelin at $EC_{50}$ (0.035 µg/mL) and $EC_{90}$ (1 µg/mL) concentrations, the expression of the DHODH gene was, respectively, decreased by 11.91% and 33.77% (p < 0.05) (*Figure 2—figure supplement 2*). This down-regulation hinders the conversion of dihydroorotate to orotate, thereby inhibiting the biosynthesis of downstream metabolites orotidine-5P and uracil, leading to their decreased abundance (*Figure 2E*). Thus, we conclude that the action mechanism of quinofumelin may involve inhibition of the de novo biosynthesis pathway of pyrimidine.

## Identification of DHODHII in *F. graminearum*

The *FgDHODHII* gene (FGSG_09678) was identified through the integrated analysis of transcriptome and metabolome data. The gene sequence for *FgDHODHII* was retrieved from the National Center for Biotechnology Information (NCBI) GenBank database. *FgDHODHII* is a gene consisting of 1419 bp without introns and is predicted to encode a protein consisting of 472 amino acids. Orthologs of *FgDHODHII* were obtained using the BLAST algorithm. The phylogenetic tree was constructed based on the amino acid sequences of DHODHII homologous proteins with Mega X using the neighbor-joining method. Motif pattern information was generated using the MEME suite (*Bailey et al., 2009*), while functional domain information and protein coordinates were obtained through CD-search (*Marchler-Bauer et al., 2009*). TBtools software was used for visualizing the tree representation (*Chen et al., 2020*). All DHODHII proteins possess the dihydroorotate dehydrogenase domain (*Figure 3*), indicating high conservation among fungi. Additionally, the amino acid sequence of the FgDHODHII exhibited 551% similarity to that of DHODHII from *Pyricularia oryzae*, as previously reported (*Higashimura et al., 2022*).

## Recovery test of mycelial growth suppressed by quinofumelin in *Fusarium genus*

The mycelial growth of *F. graminearum* PH-1 was significantly inhibited on CZA plates containing quinofumelin at concentrations of 1 µg/mL (*Figure 4*). To investigate the potential recovery of mycelial growth by exogenous dihydroorotate, uridine monophosphate (UMP), uridine, or uracil, CZA plates supplemented with quinofumelin were treated with these compounds. Despite the addition of exogenous dihydroorotate at 50 µg/mL failing to restore mycelial growth, the addition of exogenous UMP, uridine, or uracil at 50 µg/mL successfully recovered quinofumelin-induced inhibition of mycelial growth. Similar results were observed in other strains of *F. graminearum* or *F. asiaticum* (*Figure 4*; *Figure 4—figure supplement 1*). Thus, it can be concluded that quinofumelin effectively inhibits the pyrimidine biosynthesis pathways.

## The absence of *FgDHODHII* is lethal for mycelial growth in *F. graminearum*

To further investigate the biological functions of *FgDHODHII* in *F. graminearum*, we employed a homologous recombination strategy to generate target gene deletion mutants. Through PCR analysis, two putative *FgDHODHII* deletion mutants were successfully identified. The ΔFgDHODHII mutants exhibited distinct phenotypic differences compared to the wild-type progenitor PH-1 and the respective complemented strain (*Figure 5A*). These mutants displayed an inability to grow on CZA plates; however, their mycelial growth was restored when supplemented with 50 µg/mL of exogenous UMP, uridine, or uracil, while no recovery was observed with 50 µg/mL of exogenous dihydroorotate (*Figure 5B*).

## Molecular docking

To facilitate the molecular docking process between FgDHODHII and quinofumelin, ten distinct binding models were generated using AutoDockTools-1.5.7 software, each accompanied by its respective binding energies calculation results. The most favorable conformational arrangement was

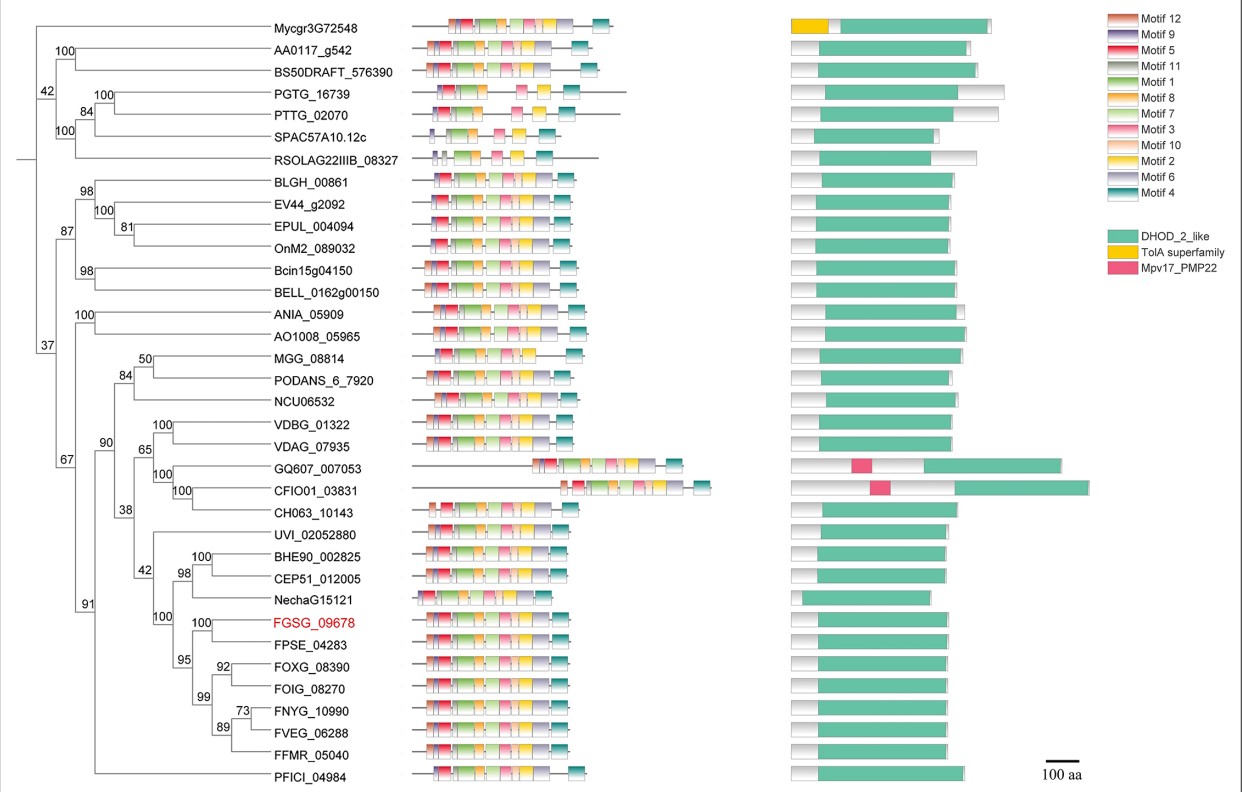

**Figure 3.** Phylogenetic tree of DHODHII proteins. The phylogenetic tree was constructed based on the amino acid sequences of DHODHII homologous proteins with Mega X using the neighbor-joining method. The bootstrap values from 1000 replications are indicated on the branches. Motif pattern information was generated using the MEME suite12, while functional domain information and protein coordinates were obtained through CD-search. The final phylogenetic tree was visualized using TBtools. FgDHODHII was highlighted in red. The amino acid sequence of *DHODHII* from *Alternaria alternata* (AA0117_g542), *Aspergillus nidulans* (ANIA_05909), *Aspergillus oryzae* (AO1008_05965), *Blumeria graminis* (BLGH_00861), *Botrytis cinerea* (Bcin15g04150), *Botrytis elliptica* (BELL_0162g00150), *Colletotrichum asianum* (GQ607_007053), *Colletotrichum fioriniae* (CFIO01_03831), *Colletotrichum higginsianum* (CH063_10143), *Corynespora cassiicola* (BS50DRAFT_576390), *Erysiphe necator* (EV44_g2092), *Erysiphe pulchra* (EPUL_004094), *Fusarium euwallaceae* (BHE90_002825), *Fusarium floridanum* (CEP51_012005), *Fusarium fujikuroi* (FFMR_05040), *Fusarium graminearum* (FGSG_09678), *Fusarium nygamai* (FNYG_10990), *Fusarium odoratissimum* (FOIG_08270), *Fusarium oxysporum* (FOXG_08390), *Fusarium pseudograminearum* (FPSE_04283), *Fusarium solani* (NechaG15121), *Fusarium verticillioides* (FVEG_06288), *Magnaporthe oryzae* (MGG_08814), *Neurospora crassa* (NCU06532), *Oidium neolycopersici* (OnM2_089032), *Pestalotiopsis fici* (PFICI_04984), *Podospora anserina* (PODANS_6_7920), *Puccinia graminis* (PGTG_16739), *Puccinia triticina* (PTTG_02070), *Rhizoctonia solani* (RSOLAG22IIIB_08327), *Schizosaccharomyces pombe* (SPAC57A10.12c), *Ustilaginoidea virens* (UVI_02052880), *Verticillium alfalfae* (VDBG_01322), *Verticillium dahliae* (VDAG_07935), and *Zymoseptoria tritici* (Mycgr3G72548) were accessed in EnsemblFungi database.

determined based on its relative ranking in terms of the computed binding energy values and subsequently visualized through PyMOL 2.5.4 software (*Figure 6A*). Notably, the obtained docking results revealed a robust interaction between FgDHODHII protein and quinofumelin as evidenced by a calculated negative free energy value of –6.8 kcal/mol.

## The precise interaction between FgDHODHII and quinofumelin was verified by surface plasmon resonance analysis

Surface plasmon resonance (SPR) was utilized to evaluate the interaction affinity between quinofumelin and FgDHODHII in this study. The experiment was conducted using a BIAcore T200 (Cytiva) instrument for biomolecular interaction analysis. FgDHODHII was immobilized as the ligand on the chip surface, while quinofumelin continuously flowed through as the analyte solution. Dilute quinofumelin to different concentration gradients (62.5, 32.15, 15.63, 7.81, 3.91 µM), measure the binding energy, and then fit the binding equilibrium dissociation constant ($K_d$) value (*Figure 6B*). The SPR results revealed that quinofumelin exhibited fast-binding and fast-dissociation modes when interacting with

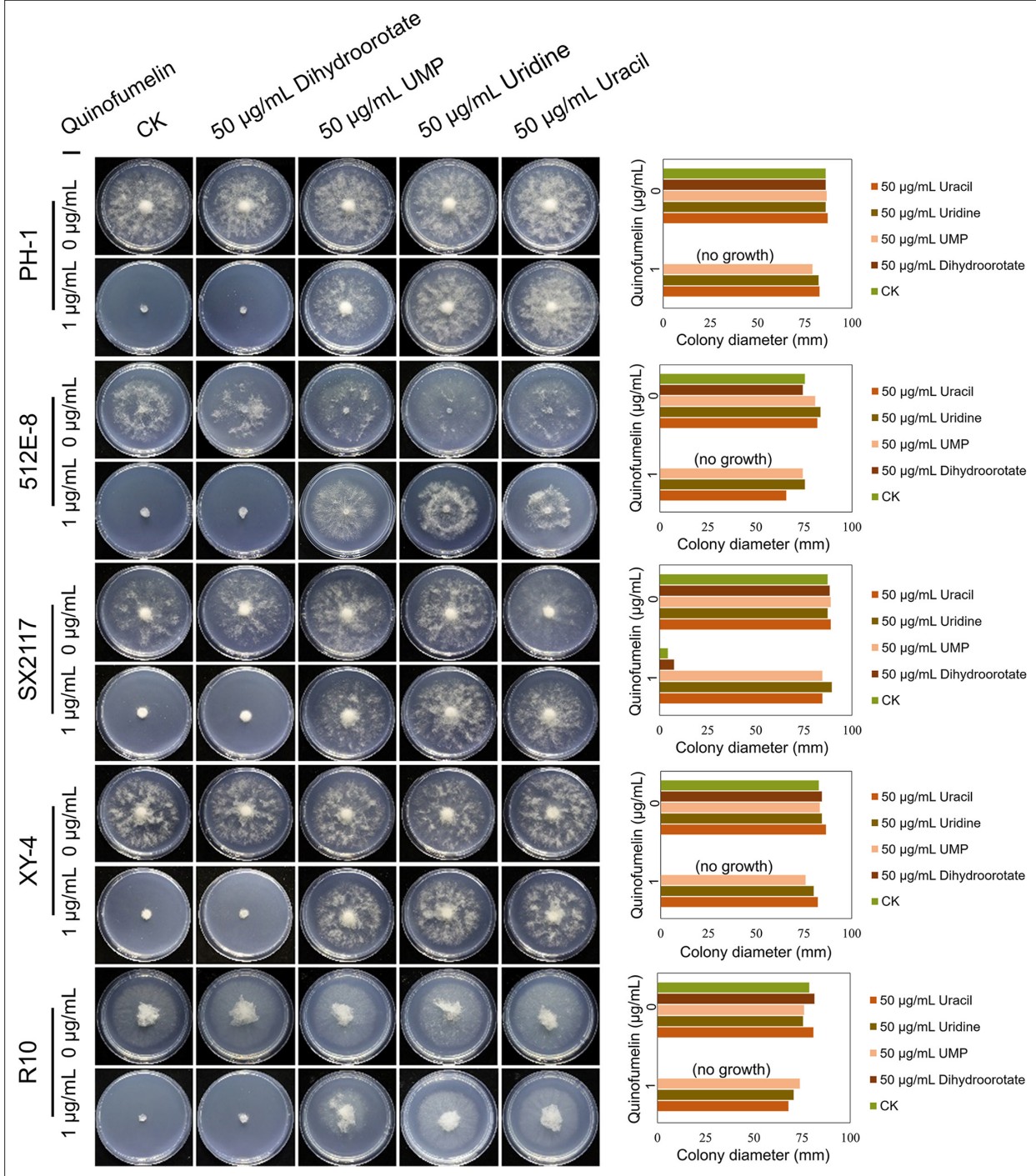

**Figure 4.** Recovery test of mycelial growth suppressed by quinofumelin. All strains were incubated on Czapek Solution Agar (CZA) plates at 25°C for 3 days. *F. graminearum* strains PH-1, 512E-8, SX2117, XY-4, and R10. The left image shows the colony morphology, while the right image is a bar chart of colony diameters.

The online version of this article includes the following figure supplement(s) for figure 4:

**Figure supplement 1.** Recovery test of mycelial growth suppressed by quinofumelin.

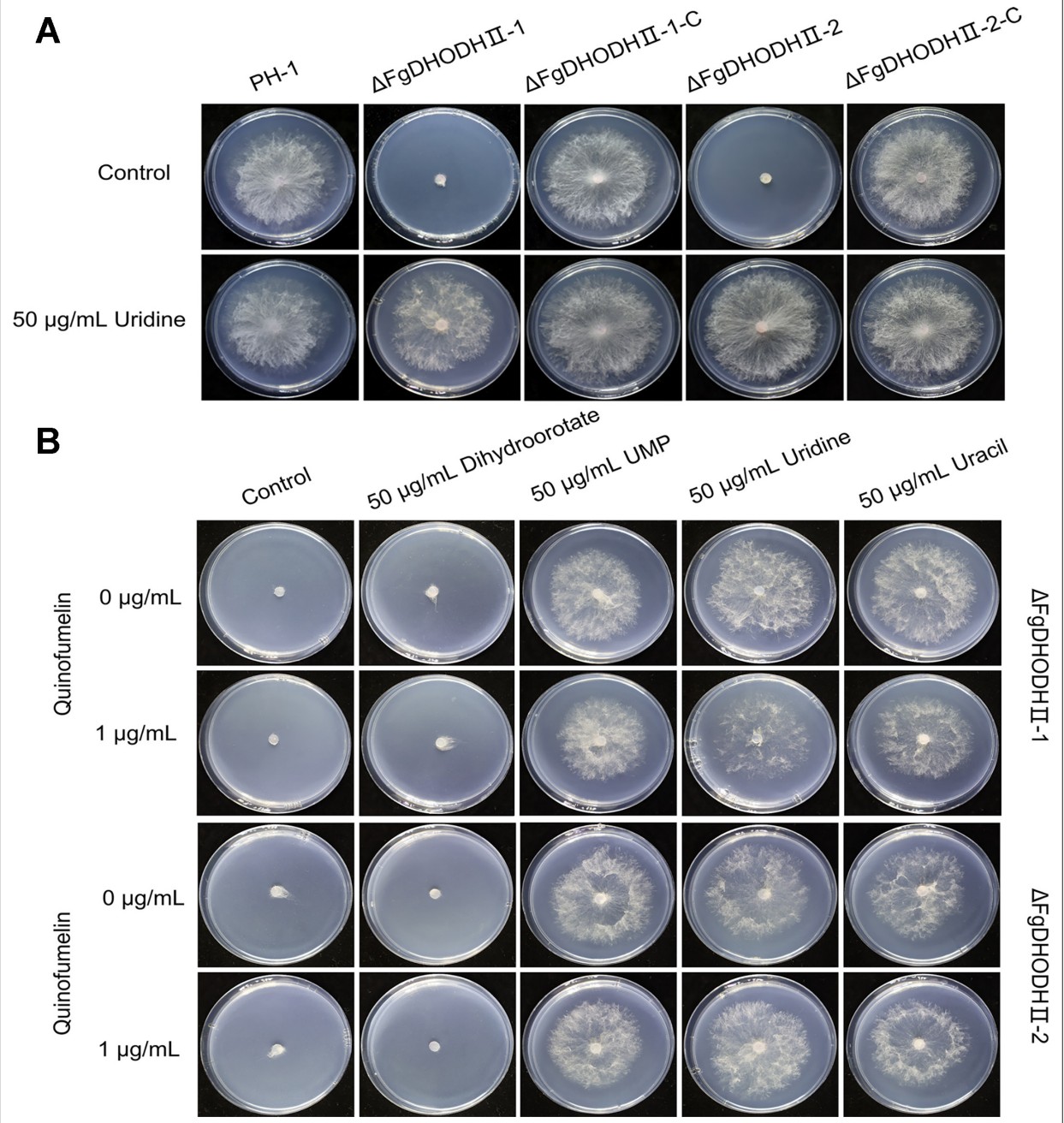

**Figure 5.** Mycelial growth of FgDHODHII deletion mutants on Czapek Solution Agar (CZA) plates. (**A**) Mycelial growth of the FgDHODHII deletion mutants and the parental strain was observed in the presence or absence of 50 µg/mL uridine. (**B**) Recovery of mycelial growth of the FgDHODHII deletion mutants was observed in the presence of 50 µg/mL dihydroorotate, uridine monophosphate (UMP), uridine, or uracil. All strains were incubated at 25°C for 3 days.

the target protein FgDHODHII. The calculated $K_D$ value was determined as $6.606 \times 10^{-6}$ M, indicating specific binding of quinofumelin to FgDHODHII.

## The interaction between FgDHODHII and quinofumelin was confirmed by microscale thermophoresis

In this study, microscale thermophoresis (MST) was further used to evaluate the interaction affinity between FgDHODHII and quinofumelin. The strength of the binding can be inferred from the Kd value. The experiment was conducted using MO.Control v1.6.1 and data analysis was performed

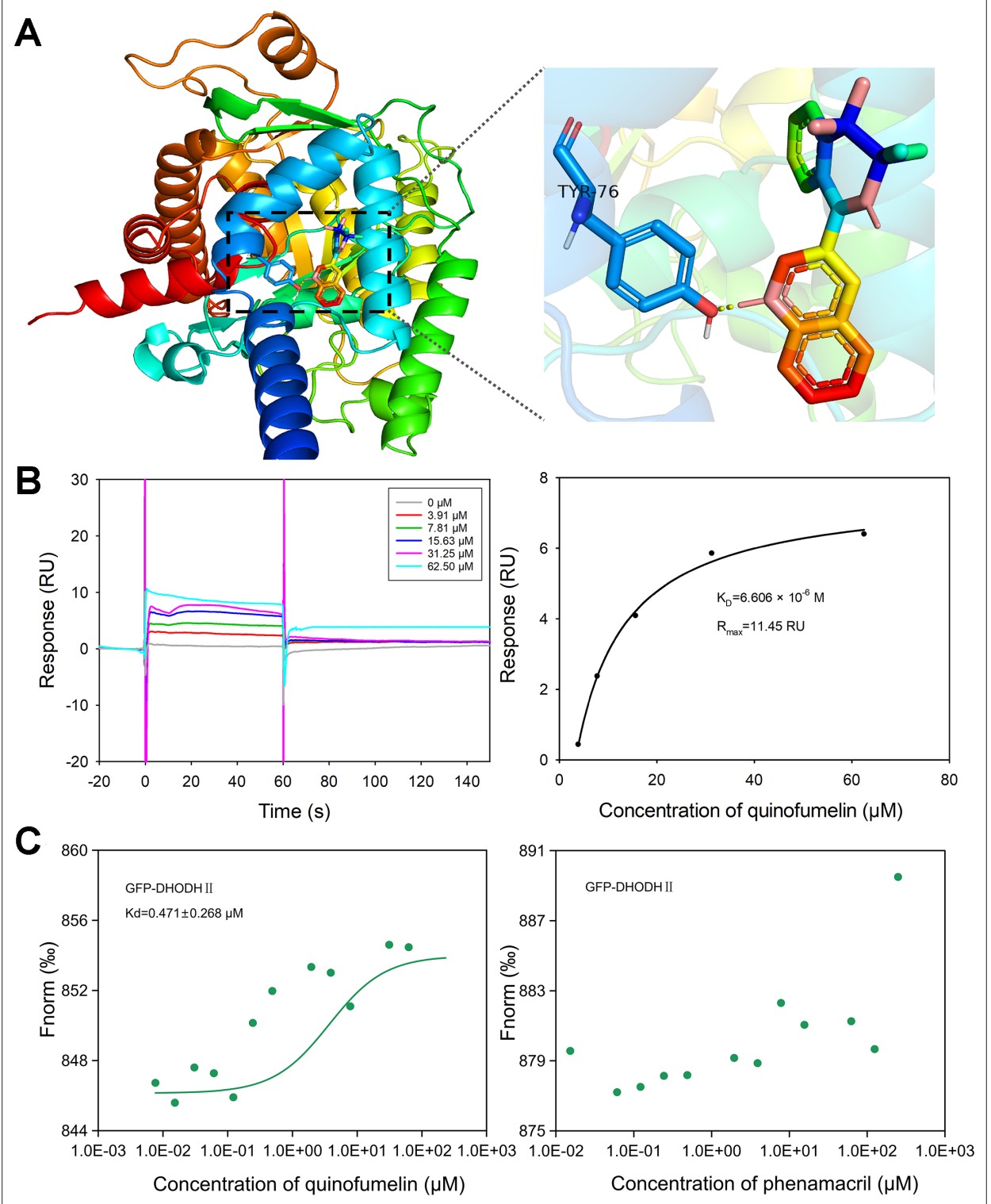

**Figure 6.** Validation and analysis of quinofumelin's binding affinity to FgDHODHII. (**A**) Three-dimensional model of the wild-type FgDHODHII are depicted, with an enlarged view highlighting the binding site for quinofumelin on the corresponding FgDHODHII models. (**B**) The surface plasmon resonance (SPR) response (RU) analysis and fitting results demonstrate the interaction between quinofumelin and FgDHODHII. (**C**) Microscale thermophoresis (MST) analysis to detect the binding of quinofumelin and FgDHODHII.

The online version of this article includes the following source data and figure supplement(s) for figure 6:

**Figure supplement 1.** SDS-PAGE electrophoresis of the fusion protein.

*Figure 6 continued on next page*

*Figure 6 continued*

**Figure supplement 1—source data 1.** TIFF file containing original western blots for *Figure 6—figure supplement 1A*, indicating the relevant bands and treatments.

**Figure supplement 1—source data 2.** Original files for SDS-PAGE analysis displayed in *Figure 6—figure supplement 1A*.

**Figure supplement 1—source data 3.** TIFF file containing original western blots for *Figure 6—figure supplement 1B*, indicating the relevant bands and treatments.

**Figure supplement 1—source data 4.** Original files for SDS-PAGE analysis displayed in *Figure 6—figure supplement 1B*.

**Figure supplement 1—source data 5.** TIFF file containing original western blots for *Figure 6—figure supplement 1C*, indicating the relevant bands and treatments.

**Figure supplement 1—source data 6.** Original files for SDS-PAGE analysis displayed in *Figure 6—figure supplement 1C*.

using MO. Affinity Analysis v2.3. By introducing a fluorescent label GFP to the FgDHODHII protein while maintaining its concentration constant, non-fluorescent labeled small molecules were diluted into 16 concentration gradients. Subsequently, the protein solution and small molecule diluent were mixed at a ratio of 19:1 before being loaded into the capillary for further analysis in the machine. The MST assay demonstrated that FgDHODHII exhibited specific binding to quinofumelin in vitro, with a dissociation constant (Kd) of 0.471±0.268 μM, whereas no binding was observed with phenamacril which lacks cross-resistance with quinofumelin (*Figure 6C*).

## Discussion

In over 130 years of research on fungicide biology, more than 30 novel targets for selective fungicides have been identified. Based on these newly discovered targets, over 200 modern selective fungicides have been developed. Consequently, the identification of novel targets for fungicides poses a significantly greater challenge compared to the development of new fungicides. Currently, modern selective fungicides with single action sites are widely applied in agricultural practices for crop disease management. However, these specific types of fungicides encounter substantial obstacles associated with resistance issues when deployed in agricultural fields. Numerous studies have extensively documented cases of plant pathogens developing resistance to modern selective fungicides, resulting in failures or reductions in their efficacy against plant diseases and subsequently leading to decreased crop yields and quality (*Duan et al., 2019*; *King et al., 2021*; *Mao et al., 2024*). For instance, *Botrytis cinerea*, a significant plant pathogenic fungus, has emerged as a multi-resistant population that is resistant to all currently used modern selective fungicides (*Shao et al., 2021*; *Weber and Petridis, 2023*). This poses a formidable challenge for the control of crop gray mold disease caused by *B. cinerea*. Therefore, there is an urgent need for the development of new fungicides with novel mechanisms of action to effectively manage multiple resistance.

The novel quinoline, quinofumelin, developed by Mitsui Chemicals Co., Ltd. Japan, has demonstrated remarkable antifungal activity against plant pathogens such as *F. graminearum* and *Sclerotinia sclerotiorum* (*Tao et al., 2021*; *Xiu et al., 2021*). Quinofumelin has been identified as an agrochemical fungicide targeting DHODHII in *Pyricularia oryzae* (*Higashimura et al., 2022*). Similarly, ipflufenoquin, another agrochemical fungicide with an analogous mechanism of action to quinofumelin, has been reported to target DHODH in *Aspergillus fumigatus* (*van Rhijn et al., 2024*). To elucidate the mode of action of quinofumelin against *F. graminearum*, we conducted a comprehensive investigation using integrated transcriptomic and metabolomic analyses. The transcriptome data revealed that DEGs were involved in 'tryptophan metabolism' and 'amino sugar and nucleotide sugar metabolism.' Furthermore, the metabolome data identified DAMs enriched in 'nucleotide metabolism' and 'amino acid metabolism.' Interestingly, we observed a co-enrichment of DEGs and DAMs in the pyrimidine metabolism pathway. Specifically, we identified a down-regulation of dihydroorotate dehydrogenase (DHODH) in the experimental groups, which impeded the conversion of dihydroorotate to orotate. Consequently, this inhibition led to decreased biosynthesis of downstream metabolites orotidine-5P and uracil. Therefore, we conducted an experiment using *F. graminearum* standard strain PH-1 to quantify mycelial growth by supplementing exogenous uracil, which is the end product of the uracil biosynthesis pathway. This was performed in the presence of complete inhibition of mycelial growth caused by quinofumelin. Surprisingly, the addition of exogenous uracil

restored the inhibited mycelial growth induced by quinofumelin (*Figure 4*). Meanwhile, similar results were successfully verified with four other *F. graminearum* strains and five *F. asiaticum* strains (*Figure 4*, *Figure 4—figure supplement 1*). Based on our depicted pyrimidine biosynthesis pathway (*Figure 2E*), we also supplemented other intermediate products, such as dihydroorotate, orotate, UMP, and uridine, externally. Except for dihydroorotate and orotate, supplementation of exogenous UMP and uridine effectively restored mycelial growth inhibited by quinofumelin as well. These findings strongly support that quinofumelin acts through modulation of the uracil biosynthesis pathway. Based on these findings and in conjunction with previous studies (*Higashimura et al., 2022*; *van Rhijn et al., 2024*), these agrochemical fungicides are categorized under FRAC group 52. This classification targets dihydroorotate dehydrogenase (DHODH) during the de novo pyrimidine biosynthesis pathway, as outlined by the Fungicide Resistance Action Committee (https://www.frac.info). As a key enzyme in the de novo pyrimidine biosynthesis pathway, DHODH has previously been investigated as a potential target for inhibiting infectious organisms and as an antitumor agent in mammals (*Chen et al., 1992*; *Mao et al., 2021*). Inhibiting DHODH can effectively impede the synthesis of DNA, RNA, and glycoproteins, thereby regulating aberrant cell metabolism and proliferation to ultimately achieve therapeutic objectives (*Boukalova et al., 2020*). Within mammalian cells, DHODH is closely associated with the mitochondrial membrane and plays a pivotal role in the respiratory chain through its interaction with ubiquinone (*Chen and Jones, 1976*). Here, using amino acid sequences from *Homo sapiens* DHODHII as a reference query, we have identified FgDHODHII (FGSG_09678), a gene comprising of 1419 bp without any introns, which encodes 472 amino acids in *F. graminearum*. FgDHODHII possesses a typical DHOD-2 like domain (NCBI: cd04738) and belongs to class 2 DHODs. Class 2 DHODHs are membrane-bound enzymes, whereas class 1 enzymes are located in the cytosol (*Nørager et al., 2002*). The DHODH enzyme catalyzes a unique redox reaction in the de novo pyrimidine biosynthesis pathway (*Dorasamy et al., 2017*). The DHOD-2-like domain can bind orotate and flavin mononucleotide (FMN), facilitating the oxidation of dihydroorotate to orotate and the reduction of the FMN cofactor (*Alves et al., 2015*). Based on the aforementioned findings, we have concluded that DHODHII functions as the target of action for quinofumelin against *F. graminearum*. Consequently, we performed a protoplast transformation assay to delete FgDHODHII in *F. graminearum* and further validate the precise target of action for quinofumelin. Typically, most

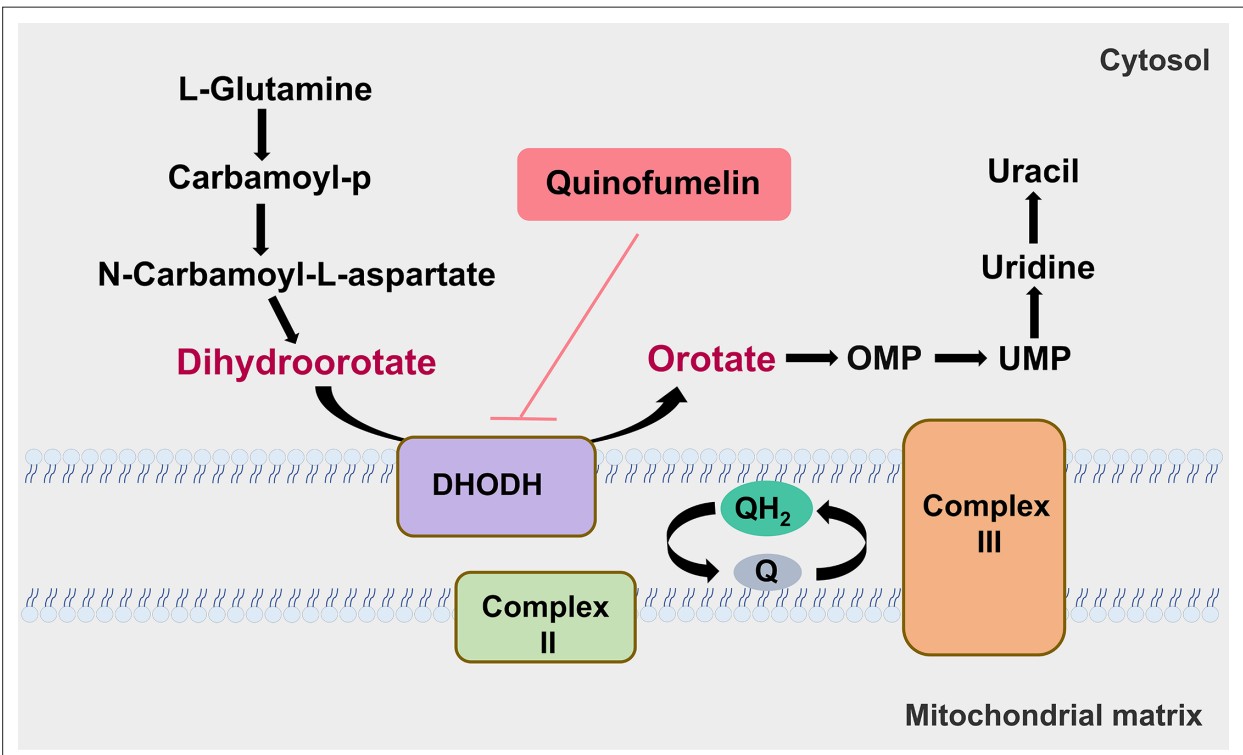

**Figure 7.** A model of the pyrimidine pathway that quinofumelin attacks.

fungicide targets are either lethal genes or essential pathogenic genes. Therefore, it is plausible that the inactivation of FgDHODHII may be lethal. In our genetic transformation experiment, exogenous uridine was supplemented to successfully obtain deletion mutants of the FgDHODHII gene. Fortunately, two ΔFgDHODHII mutants were successfully obtained. These mutants exhibited an inability to grow on PDA or CZA medium; however, mycelial growth was fully restored upon addition of uridine to the medium (*Figure 5A*). Additionally, exogenous supplementation of UMP or uracil effectively restored mycelial growth. However, the supplementation of dihydroorotate to the medium did not alleviate the growth defect observed in ΔFgDHODHII mutants (*Figure 5B*). Furthermore, mycelial growth in two ΔFgDHODHII mutants was restored by exogenously adding UMP, uracil or uridine under quinofumelin treatments. These findings indicate that the defects of FgDHODHII are functionally equivalent to complete inhibition of its activity by quinofumelin and further confirm that the mechanism of action for quinofumelin involves de novo synthesis of uracil while identifying FgDHODHII as its specific target (*Figure 7*).

Molecular docking is a computational procedure employed to predict the optimal binding orientation of a ligand with its macromolecular target (receptor), thereby forming a stable complex. In concise terms, docking is a molecular modeling technique employed to predict the interaction between a protein (enzyme) and small molecules (ligands). In this study, we conducted molecular docking analysis between quinofumelin and the target protein FgDHODHII. The docking results unveiled a robust affinity between FgDHODHII and quinofumelin. Furthermore, these findings provided additional evidence that the strong affinity of quinofumelin with FgDHODHII disrupts its activity and function.

SPR technique is employed for real-time measurement of molecular interactions (*Olaru et al., 2015*). The SPR sensor map characterizes intermolecular interaction rates primarily by conveying the kinetic rate constant, while it characterizes the binding level between molecules mainly by conveying the affinity constant (*Guo, 2012*; *Piliarik et al., 2009*). In this study, we used SPR technology to measure the binding energy and kinetics between FgDHODHII and quinofumelin. The binding kinetics of FgDHODHII and quinofumelin demonstrated fast-binding and fast-dissociation modes. Soluble FgDHODHII exhibited an ideal affinity (Kd) of $6.606 \times 10^{-6}$ M with quinofumelin, falling within the suitable range of $10^{-3}$ M to $10^{-6}$ M for biomacromolecule-small molecule interactions. Furthermore, the binding affinity between FgDHODHII and quinofumelin was confirmed by a microscale thermophoresis (MST) assay. Therefore, these data strongly support the precise interaction between quinofumelin and its target protein FgDHODHII. This newly discovered selective target for antimicrobial agents offers a valuable resource for the design and development of targeted pesticides.

Unlike many other dehydrogenases that require NAD or NADP as an electron acceptor, DHODH transfers electrons to ubiquinone, which then transfers electrons into respiratory complex III and eventually to oxygen (*Wang and Hekimi, 2016*). Thus, the production of orotate in pyrimidine biosynthesis is coupled with the electron transport system and energy production, exerting additional control over nucleotide synthesis and ensuring its coupling to cellular energy metabolism. In the pyrimidine biosynthesis pathway, DHODHII regulates the conversion from dihydroorotate to orotate. Our findings demonstrated that DHODHII is a target of action for quinofumelin. A previous study reported that exogenous orotate but not exogenous dihydroorotate recovered mycelial growth inhibited by quinofumelin, indicating that quinofumelin inhibits the formation of orotate in the pyrimidine biosynthesis pathway (*Higashimura et al., 2022*). However, we observed that the supplementation of orotate to the medium did not alleviate the growth defect of ΔFgDHODHII mutants. Considering the disparity in mycelial growth inhibition between *F. graminearum* and other fungi, our data suggest that a nonclassical two-site ping-pong mechanism is the most suitable model for DHODH as previously reported (*Chen et al., 1992*). In the classical ping-pong mechanism, DHODH interacts with dihydroorotate and ubiquinol in an identical conformation, while orotate and ubiquinone combine with an alternative form (*Cleland, 1963*). However, similar to brequinar sodium inhibiting the activity of DHODH isolated from mammalian sources, FgDHODHII may follow a nonclassical, two-site ping-pong mechanism. In this non-classical ping-pong mechanism, the two substrates may not share chemical resemblance and, therefore, do not bind to a single binding site but likely bind to two adjacent sites that facilitate reactant transfer. The precise nature of the interaction between quinofumelin and FgDHODHII may require further elucidation studies.

## Materials and methods

### Strains, culture conditions, and fungicides

*F. graminearum* sensitive strain PH-1 and 9 wild-type strains (*F. graminearum* and *F. asiaticum*), and the PH-1 deletion mutants ΔFgDHODHII were maintained in the Fungicide Biology Laboratory, Nanjing Agricultural University.

The media used in this study included potato dextrose agar (PDA) medium, Yeast extract peptone dextrose (YEPD) medium, Czapek Solution Agar (CZA) medium, and LB medium.

Chemical-grade quinofumelin (Mitsui Agricultural Chemical Company of Japan) was prepared with dimethyl sulfoxide at a concentration of $1 \times 10^4$ µg/mL and stored at 4°C.

### Transcriptomic assay of mycelial growth in *F. graminearum* as affected by quinofumelin

A total of 20 mL of YEPD medium containing 1 mL of conidia suspension ($1 \times 10^5$ conidia/mL) was incubated with shaking (175 rpm/min) at 25°C. After 24 hr, the medium was added with quinofumelin ($EC_{90}$) at a concentration of 1 µg/mL, while an equal amount of dimethyl sulfoxide was added as the control (CK). The incubation continued for another 48 hr, followed by filtration and collection of hyphae. Subsequently, Genedenovo Biotechnology Co., Ltd. (Guangzhou, China) performed sequencing on the Illumina platform. Quality control measures were implemented to remove linker sequences and low-quality bases from the original reads in order to obtain clean data readings. The filtered data underwent analysis for base composition and mass distribution to visualize data quality. Based on the gene expression data, we used R (http://www.r-project.org/) for conducting principal component analysis, while the 'pheatmap' package was used to generate the clustering heatmap. DESeq2 was applied to analyze group differences based on quantitative expression results. The screening threshold of p-value <0.05 and FC ≥1.5 or FC ≤0.67 was used to identify unigenes with significant differential expression between the two groups using a t-test. Finally, functional enrichment analysis of GO and KEGG pathways was performed on these differentially expressed unigenes.

### Metabonomic assay of mycelial growth in *F. graminearum* as affected by quinofumelin

Transcriptome and metabolome samples were obtained from the same treatment group. The widely targeted metabolome analysis was conducted by Metware Biotechnology Co., Ltd. (Wuhan, China). The frozen sample stored at –80°C was thawed on ice. A 400 µL solution (Methanol: Water = 7:3, v/v) containing an internal standard was added to a 20 mg sample and vortexed for 3 min. Subsequently, the sample was sonicated in an ice bath for 10 min and vortexed for 1 min before being placed at –20°C for 30 min. After centrifugation at 12,000 rpm for 10 min (4°C). The sediment was removed, followed by another centrifugation of the supernatant at 12,000 rpm for 3 min (4°C). A volume of 200 µL aliquots of supernatant was transferred for LC-MS analysis. All samples were acquired using the LC-MS system according to machine orders. The original data file acquired by LC-MS was converted into mzML format using ProteoWizard software. Peak extraction, peak alignment, and retention time correction were performed by XCMS program. The identified metabolites were analyzed using MBEole2.0 (csbg.cnb.csic.es/mbrole2/index.php) and KEGG (http://www.kegg.jp). PLS-DA analysis was used to identify the differentially accumulated metabolites (DAMs), with a screening threshold of p-value <0.05 and FC ≥1.5 or FC ≤0.67. Each sample was replicated six times for biological consistency.

### Recovery assay of quinofumelin-induced mycelial growth inhibition in *F. graminearum*

*F. graminearum* sensitive strain PH-1 and 9 wild-type strains (*F. graminearum* and *F. asiaticum*), and the PH-1 deletion mutants ΔFgDHODHII, were preincubated on a PDA plate to form mycelial colonies. Mycelial plugs (5 mm in diameter) from the edge of a 3-day-old colony were removed and inoculated onto a CZA plate containing quinofumelin (1 µg/mL) and dihydroorotate, UMP, uridine, or uracil at 50 µg/mL, followed by measurement of colony diameters. Three replicates were conducted for each concentration.

### Construction of deletion mutant vectors

The deletion mutant of *FgDHODHII* was generated through protoplast transformation, as previously described with some modifications in the transformation procedures (*Zhang et al., 2016*). For the

transformation process, washed protoplasts were suspended in 160 µL sorbitol-tris-calcium chloride (STC) solution and 40 µL sorbitol-peg-tris-calcium chloride (SPTC) solution (40% polyethylene glycol 6000 in STC), along with 2 µg of the aforementioned DNA fragments and 10 µL of heparin (5 mg/mL in STC) to form protoplast suspensions. Subsequently, these mixtures were incubated on ice for 30 min, followed by mixing with 1 mL of SPTC and further incubation at room temperature for 20 min. The resulting suspensions were gently mixed into regeneration medium (RM) (100 mL) containing 50 µg/mL uridine and poured into plates. After incubating for 24 hr at 25°C, the plates were overlaid with selective regeneration medium (SRM) containing 50 µg/mL uridine and continuously incubated for an additional 4 days. The putative transformants could be observed growing on SRM. All primers used are listed in *Table 1*.

## Molecular docking

The binding affinity of quinofumelin with FgDHODHII protein was assessed through molecular docking using AutoDockTools-1.5.7 software. The three-dimensional structure of quinofumelin (PubChem CID: 23160856) used for docking was obtained from PubChem database, while the three-dimensional structural model of *F. graminearum* DHODHII protein predicted by AlphaFold (*Jumper et al., 2021*) was obtained from the UniProt database (*Bateman et al., 2019*). In the docking studies, binding energy was employed as a metric to evaluate the strength of protein-ligand interaction, where lower values indicated higher affinity between proteins and ligands. The resulting docking outcomes were visualized using PyMOL software version 2.5.4.

## SPR verification of specific interactions between FgDHODHII and quinofumelin

The cDNA sequence of FgDHODHII was analyzed and codon optimization was performed for the *E. coli* expression system (*Kyriakopoulos and Kontoravdi, 2013*; *Yang et al., 2021*). Subsequently, it was ligated into the pCold-9×his MBP plasmid and the pET-28a plasmid at the NdeI and EcoRI sites within the multiple cloning sites to construct recombinant plasmids: pCold-9×his-MBP-TEV-DHODHII

**Table 1.** Primers used in the study.

| Primer | Sequence (5'–3') | Use |
|---|---|---|
| FgDHODHII-UF | GGCAGTGAAAGTCTTGTTCAAG | |
| FgDHODHII-UR | GCTCCTTCAATATCATCTTCTGTG GCTGGGAAGTAATGCTGGG | PCR primers to amplify FgDHODHII upstream fragment |
| FgDHODHII-DF | GAGACAATACCGGAAGGAAC ACATGCTATAGCAAGCAACA | |
| FgDHODHII-DR | AAACTTAACAAATCATCTTTGCC | PCR primers to amplify FgDHODHII downstream fragment |
| SS-F | ACAGAAGATGATATTGAAGGAGC | |
| SS-R | GAGACAATACCGGAAGGAA | PCR primers to amplify Hph-Hsv-tk fragment |
| FgDHODHII-YZ-F | TTTGCCCGATCTCTTGGTCT | |
| FgDHODHII-YZ-R | AACCACCAGTCTCCATGAGG | PCR primers to verify FgDHODHII |
| FgDHODHII-GFP-F | CTCATCACCATCACCATCAC ATGTCTGCCGCTCTCCTACG | |
| FgDHODHII-GFP-R | AGCTCCTCGCCCTTGCTCAC CCTCGAATTCTTCACACCAC | Constructed FgDHODHII-GFP fusion vector |
| T7 | TAATACGACTCACTATAGG | |
| T7t | CCGCTAGTTATTGCTCAGC | Amplification of FgDHODHII fragments for sequencing |
| pET-28a-F | CCTGGTGCCGCGCGGCAGCCATATGCT GGCTGTCAATGTTTTTGGAACAG | |
| pET-28a-R | AAGCTTGTCGACGGAGCTCGAATTCCT ATTTTAACTCCTGCTTGATCCTG | PCR primers to construct pET28a-FgDHODHII expression vector |
| pCold-F | GTAAAGCACGCCATATCGC | |
| pCold-R | CCAAATGGCAGGGATCTTAG | PCR primers to construct pCold-FgDHODHII expression vector |

(*Fang et al., 2018*), pET-28a (+)-DHODHII and pET-28a (+)-DHODHII[124-467] (with removal of insoluble fragments) (*Figure 6—figure supplement 1*).

Using primers T7 and T7t for sequencing resulted in the identification of the correct transformants. Plasmid DNA was extracted using the E.Z.N.A Plasmid Extraction DNA Mini Kit I protocol (Omega, China). For protein overexpression, the extracted plasmids were separately transformed into *E. coli* BL21 strain (*Zhu et al., 2014*). A single clone obtained from each transformation was used for protein overexpression and purification. Upon reaching an OD600 of 1.0, 0.6 mM isopropyl-β-D-thiogalactopyranoside (IPTG) was added to induce protein overexpression. Following induction at 16°C for 9 hr, cells were harvested, lysed, and purified using a Ni-NTA column (QIAGEN).

The interaction between FgDHODHII and quinofumelin was evaluated using a Biacore T200 instrument (GE Healthcare). The CM7 sensor chip was employed for immobilizing 50 μg/mL of purified FgDHODHII protein via amino coupling method at a flow rate of 10 μL/min, utilizing 1×PBS-P+containing 5% DMSO as running buffer. The desired coupling amount of FgDHODHII protein on the chip surface was targeted to be within the range of 12,000–24,000 RU. To investigate binding affinity, quinofumelin concentrations were diluted from 3.90625 to 62.5 μM in a constant flow rate running buffer containing 5% DMSO. Following sample injection, a switching buffer was introduced to allow spontaneous dissociation of quinofumelin from FgDHODHII on the chip surface, and this dissociation process was continuously monitored in real time by measuring response values. The affinity value (Kd) was determined through kinetic analysis or steady-state affinity method.

## Identification of specific interactions between FgDHODHII and quinofumelin using MST

To construct the FgDHODHII-GFP cassette, the FgDHODHII fragment containing the native promoter and ORF (without the stop codon) was amplified. Subsequently, the resulting PCR product was co-transformed with XhoI-digested pYF11 into *E. coli* DH5α as previously reported (*Tang et al., 2020*). The fusion vector was then used to transform PH-1 to obtain the FgDHODHII-GFP strain. The GFP fusion protein was extracted from mycelia. Two Monolith NT.115 standard capillaries were employed for repeated absorption of a specific protein concentration, and placed in positions 1 and 2 of the sample tray in the MonoTemper Monolith NT.115 instrument. Sixteen PCR tubes were prepared, with 10 μL DMSO added to PCR tubes 2–16 and 20 μL of 5000 μM quinofumelin (or phenamacril) dissolved in DMSO added to PCR tube 1. Then, 10 μL of quinofumelin from PCR tube 1 was transferred and thoroughly mixed with PCR tube 2 using a pipette. These steps were repeated for PCR tubes 3–16. Following mixing with PCR tube 16 (discarding the remaining 10 μL), each PCR tube was thoroughly mixed with 190 μL DHODHII-GFP protein, and incubated at room temperature for 30 min. Monolith NT.115 standard capillaries were inserted into each PCR tube to allow solution entry into the capillary. The capillaries 1–16 were subsequently placed in the sample tray of the device and the program was initiated. All measurements were performed using 80% (Auto-detect) Nano-RED. MonoTemper Monolith NT.115 software was used to evaluate the binding affinity between the target protein and quinofumelin using the Kd values and the signal-to-noise ratio, and the results were combined and analyzed using Mo. Affinity Analysis v2.3 software (*Rotem et al., 2016*).

### Statistical analysis

The data in the study were analyzed using analysis of variance (ANOVA) with the SPSS 14.0 software (SPSS lnc, Chicago, IL, USA). When ANOVA yielded a significant result ($p=0.05$), means were compared using Fisher's protected least significant difference (PLSD).

## Acknowledgements

This work was supported by the National Key Research and Development Program of China (2022YFD1400100), and the National Natural Science Foundation of China (32072448).

## Additional information

### Funding

| Funder | Grant reference number | Author |
|---|---|---|
| National Key Research and Development Program of China | 2022YFD1400100 | Yabing Duan |
| National Natural Science Foundation of China | 32072448 | Mingguo Zhou |

The funders had no role in study design, data collection and interpretation, or the decision to submit the work for publication.

### Author contributions

Qian Xiu, Data curation, Software, Investigation, Methodology, Writing - original draft; Xiaoru Yin, Data curation, Software, Investigation, Methodology; Yuanyuan Chen, Data curation, Software, Investigation; Ziyang Zhang, Data curation, Investigation, Writing – review and editing; Yushuai Mao, Data curation, Methodology; Tianshi Wang, Software; Jie Zhang, Supervision, Methodology; Mingguo Zhou, Conceptualization, Supervision, Funding acquisition; Yabing Duan, Conceptualization, Supervision, Funding acquisition, Project administration, Writing – review and editing

### Author ORCIDs

Xiaoru Yin ⓘ https://orcid.org/0009-0007-2877-6950
Ziyang Zhang ⓘ https://orcid.org/0009-0004-4779-3041
Jie Zhang ⓘ https://orcid.org/0009-0001-0190-7574
Yabing Duan ⓘ https://orcid.org/0000-0002-4183-8729

Reviewer #2 (Public review): https://doi.org/10.7554/eLife.105892.3.sa1
Reviewer #3 (Public review): https://doi.org/10.7554/eLife.105892.3.sa2
Author response https://doi.org/10.7554/eLife.105892.3.sa3

## Additional files

### Supplementary files

MDAR checklist

Supplementary file 1. Supplementary tables for this article.

### Data availability

All datasets generated during this study have been deposited in the Dryad Digital Repository and are publicly available under the https://doi.org/10.5061/dryad.n5tb2rc7r.

The following dataset was generated:

| Author(s) | Year | Dataset title | Dataset URL | Database and Identifier |
|---|---|---|---|---|
| Xiu Q, Yin XR, Chen YY, Zhang ZY, Mao YS, Wang TS, Zhang J, Zhou MG, Duan YB | 2025 | Action mechanism of a novel agrichemical quinofumelin against Fusarium graminearum | https://doi.org/10.5061/dryad.n5tb2rc7r | Dryad Digital Repository, 10.5061/dryad.n5tb2rc7r |

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
