## [Editor Report · eLife Assessment]

In this **valuable** study, the authors show the physiological response and molecular pathway mediating the effect of quinofumelin, a developed fungicide with an unknown mechanism. The authors present **convincing** data suggesting the involvement of the uridine/uracil biosynthesis pathway, by combining in vivo microbiology characterization as well as in vitro biochemical binding results.

---

## [Referee Report · Reviewer #2 (Public review)]

Summary:

In the current study, the authors aim to identify the mode of action/molecular mechanism of characterized a fungicide, quinofumelin, and its biological impact on transcriptomics and metabolomics in Fusarium graminearum and other Fusarium species. Two sets of data were generated between quinofumelin and no treatment group, and differentially abundant transcripts and metabolites were identified, suggesting a potential role of pyrimidine biosynthesis. Upon studying the genetic mutants of the uridine/uracil biosynthesis pathway with quinofumelin treatment and metabolite supplementation, combining in vitro biochemical assay of quinofumelin and F.graminearum dihydroorotate dehydrogenase protein, the authors identified that quinofumelin inhibits the dihydroorotate dehydrogenase and blocks downstream metabolite biosynthesis, limiting fungal metabolism and growth.

Strengths:

Omics datasets were leveraged to understand the physiological impact of quinofumelin, showing the intracellular impact of the fungicide. The characterization of FgDHODHII deletion strains with supplemented metabolites clearly showed the impact of the enzyme on fungal growth. Corroborating in vitro and in vivo data revealed the direct interaction of quinofumelin with Fusarium protein target.

Potential Impact:

Understanding this new mechanism could facilitate rational design or screen for molecules targeting the same pathway, or improve binding affinity and inhibitor potency. Confirming the target of quinofumelin may also help understand its resistance mechanism, and further development of other inhibitory molecules against the target.

---

## [Referee Report · Reviewer #3 (Public review)]

Summary:

The manuscript shows the mechanism of action of quinofumelin, a novel fungicide, against the fungus Fusarium graminearum. Through omics analysis, phenotypic analysis and in silico approaches, the role of quinofumelin in targeting DHODH is uncovered.

Strengths:

The phenotypic analysis and mutant generation are nice data and add to the role of metabolites in bypassing pyrimidine biosynthesis.

Weaknesses:

The role of DHODH in this class of fungicides has been known and this data does not add any further significance to the field.

---

## [Author Response]

The following is the authors’ response to the original reviews.

**Reviewer #1 (Public review):**
Summary:Phytophathogens including fungal pathogens such as *F. graminearum* remain a major threat to agriculture and food security. Several agriculturally relevant fungicides including the potent Quinofumelin have been discovered to date, yet the mechanisms of their action and specific targets within the cell remain unclear. This paper sets out to contribute to addressing these outstanding questions.

We appreciate the reviewer's accurate summary of our manuscript.

Strengths:The paper is generally well-written and provides convincing data to support their claims for the impact of Quinofumelin on fungal growth, the target of the drug, and the potential mechanism. Critically the authors identify an important pyrimidine pathway dihydroorotate dehydrogenase (DHODH) gene FgDHODHII in the pathway or mechanism of the drug from the prominent plant pathogen *F. graminearum*, confirming it as the target for Quinofumelin. The evidence is supported by transcriptomic, metabolomic as well as MST, SPR, molecular docking/structural biology analyses.

We appreciate the reviewer's recognition of the strengths of our manuscript.

Weaknesses:Whilst the study adds to our knowledge about this drug, it is, however, worth stating that previous reports (although in different organisms) by Higashimura et al., 2022 https://pmc.ncbi.nlm.nih.gov/articles/PMC9716045/ had already identified DHODH as the target for Quinofumelin and hence this knowledge is not new and hence the authors may want to tone down the claim that they discovered this mechanism and also give sufficient credit to the previous authors work at the start of the write-up in the introduction section rather than in passing as they did with reference 25? other specific recommendations to improve the text are provided in the recommendations for authors section below.

We appreciate the reviewer's suggestion. In the revised manuscript, we have incorporated the reference in the introduction section and expanded the discussion of previous work on quinofumelin by Higashimura et al., 2022 in the discussion section to more effectively contextualize their contributions. Moreover, we have made revisions and provided responses in accordance with the recommendations.

**Reviewer #2 (Public review):**
Summary:In the current study, the authors aim to identify the mode of action/molecular mechanism of characterized a fungicide, quinofumelin, and its biological impact on transcriptomics and metabolomics in Fusarium graminearum and other Fusarium species. Two sets of data were generated between quinofumelin and no treatment group, and differentially abundant transcripts and metabolites were identified. The authors further focused on uridine/uracil biosynthesis pathway, considering the significant up- and down-regulation observed in final metabolites and some of the genes in the pathways. Using a deletion mutant of one of the genes and in vitro biochemical assays, the authors concluded that quinofumelin binds to the dihydroorotate dehydrogenase.

We appreciate the reviewer's accurate summary of our manuscript.

Strengths:Omics datasets were leveraged to understand the physiological impact of quinofumelin, showing the intracellular impact of the fungicide. The characterization of FgDHODHII deletion strains with supplemented metabolites clearly showed the impact of the enzyme on fungal growth.

We appreciate the reviewer's recognition of the strengths of our manuscript.

Weaknesses:Some interpretation of results is not accurate and some experiments lack controls. The comparison between quinofumelin-treated deletion strains, in the presence of different metabolites didn't suggest the fungicide is FgDHODHII specific. A wild type is required in this experiment.Potential Impact: Confirming the target of quinofumelin may help understand its resistance mehchanism, and further development of other inhibitory molecules against the target.The manuscript would benefit more in explaining the study rationale if more background on previous characterization of this fungicide on Fusarium is given.

We appreciate the reviewer's suggestion. Under no treatment with quinofumelin, mycelial growth remains normal and does not require restoration. In the presence of quinofumelin treatment, the supplementation of downstream metabolites in the de novo pyrimidine biosynthesis pathway can restore mycelial growth that is inhibited by quinofumelin. The wild-type control group is illustrated in Figure 4. Figure 5b depicts the phenotypes of the deletion mutants. With respect to the relationship among quinofumelin, FgDHODHII, and other metabolites, quinofumelin specifically targets the key enzyme FgDHODHII in the de novo pyrimidine biosynthesis pathway, disrupting the conversion of dihydroorotate to orotate, which consequently inhibits the synthesis downstream metabolites including uracil. In our previous study, quinofumelin not only exhibited excellent antifungal activity against the mycelial growth and spore germination of *F. graminearum*, but also inhibited the biosynthesis of deoxynivalenol (DON). We have added this part to the introduction section.

**Reviewer #3 (Public review):**
Summary:The manuscript shows the mechanism of action of quinofumelin, a novel fungicide, against the fungus Fusarium graminearum. Through omics analysis, phenotypic analysis, and in silico approaches, the role of quinofumelin in targeting DHODH is uncovered.

We appreciate the reviewer's accurate summary of our manuscript.

Strengths:The phenotypic analysis and mutant generation are nice data and add to the role of metabolites in bypassing pyrimidine biosynthesis.

We appreciate the reviewer's recognition of the strengths of our manuscript.

Weaknesses:The role of DHODH in this class of fungicides has been known and this data does not add any further significance to the field. The work of Higashimura et al is not appreciated well enough as they already showed the role of quinofumelin upon DHODH II.There is no mention of the other fungicide within this class ipflufenoquin, as there is ample data on this molecule.

We appreciate the reviewer's suggestion. We sincerely appreciate the reviewer's insightful comment regarding the work of Higashimura et al. We agree that their investigation into the role of quinofumelin in DHODH II inhibition provides critical foundational insights for this field. In the revised manuscript, we have incorporated the reference in the introduction section and expanded the discussion of their work in the discussion section to more effectively contextualize their contributions. The information regarding action mechanism of ipflufenoquin against filamentous fungi was added in discussion section.

**Reviewer #1 (Recommendations for the authors):**
(1) Given that the DHODH gene had been identified as a target earlier, could the authors perform blast experiments with this gene instead and let us know the percentage similarity between the FgDHODHII gene and the Pyricularia oryzae class II DHODH gene in the report by Higashimura et al., 2022.

BLAST experiment revealed that the percentage similarity between the FgDHODHII gene and the class II DHODH gene of *P. oryzae* was 55.41%. We have added the description ‘Additionally, the amino acid sequence of the FgDHODHII exhibits 55.41% similarity to that of DHODHII from *Pyricularia oryzae*, as previously reported (Higashimura et al., 2022)’ in section Results.

(2) Abstract:The authors started abbreviating new terms e.g. DEG, DMP, etc but then all of a sudden stopped and introduced UMP with no full meaning of the abbreviation. Please give the full meaning of all abbreviations in the text, UMP, STC, RM, etc.

We have provided the full meaning for all abbreviations as requested.

(3) Introduction section:The introduction talks very little about the work of other groups on quinofumelin. Perhaps add this information in and reference them including the work of Higashimura et al., 2022 which has done quite significant work on this topic but is not even mentioned in the background

We have added the work of other groups on quinofumelin in section introduction.

(4) General statements:Please show a model of the pyrimidine pathway that quinofumelin attacks to make it easier for the reader to understand the context. They could just copy this from KEGG

We have added the model (Fig. 7).

(5) Line 186:The authors did a great job of demonstrating interactions with the Quinofumelin and went to lengths to perform MST, SPR, molecular docking, and structural biology analyses yet in the end provide no details about the specific amino acid residues involved in the interaction. I would suggest that site-directed mutagenesis studies be performed on FgDHODHII to identify specific amino acid residues that interact with Quinofumelin and show that their disruption weakens Quinofumelin interaction with FgDHODHII.

Thank you for this insightful suggestion. We fully agree with the importance of elucidating the interaction mechanism. At present, we are conducting site-directed mutagenesis studies based on interaction sites from docking results and the mutation sites of FgDHODHII from the resistant mutants; however, due to the limitations in the accuracy of existing predictive models, this work remains ongoing. Additionally, we are undertaking co-crystallization experiments of FgDHODHII with quinofumelin to directly and precisely reveal their interaction pattern

(6) Line 76:What is the reference or evidence for the statement 'In addition, quinofumelin exhibits no cross-resistance to currently extensively used fungicides, indicating its unique action target against phytopathogenic fungi.

If two fungicides share the same mechanism of action, they will exhibit cross resistance. Previous studies have demonstrated that quinofumelin retains effective antifungal activity against fungal strains resistant to commercial fungicides, indicating that quinofumelin does not exhibit cross-resistance with other commercially available fungicides and possesses a novel mechanism of action. Additionally, we have added the relevant inference.

(7) Line 80-82:Again, considering the work of previous authors, this target is not newly discovered. Please consider toning down this statement 'This newly discovered selective target for antimicrobial agents provides a valuable resource for the design and development of targeted pesticides.'

We have rewritten the description of this sentence.

(8) Line 138: If the authors have identified DHODH in experimental groups (I assume in F. graminearum), what was the exact locus tag or gene name in F. graminearum, and why not just continue with this gene you identified or what is the point of doing a blast again to find the gene if the DHODH gene if it already came up in your transcriptomic or metabolic studies? This unfortunately doesn't make sense but could be explained better.

The information of *FgDHODHII* (gene ID: FGSG_09678) has been added. We have revised this part.

**Reviewer #2 (Recommendations for the authors):**
(1) Line 40:Please add a reference.

We have added the reference

(2) Line 47:Please add a reference.

We have added the reference.

(3) Line 50:The lack of target diversity in existing fungicides doesn't necessarily serve as a reason for discovering new targets being more challenging than identifying new fungicides within existing categories, please consider adjusting the argument here. Instead, the authors can consider reasons for the lack of new targets in the field.

We have revised the description.

(4) Line 63:Please cite your source with the new technology.

We have added the reference.

(5) Line 68:What are you referring to for "targeted medicine", do you have a reference?

We have revised the description and the reference.

(6) Line 74:One of the papers referred to "quinoxyfen", what are the similarities and differences between the two? Please elaborate for the readership.

Quinoxyfen, similar to quinofumelin, contains a quinoline ring structure. It inhibits mycelial growth by disrupting the MAP kinase signaling pathway in fungi (https://www.frac.info). In addition, quinoxyfen still exhibits excellent antifungal activity against the quinofumelin-resistant mutants (the findings from our group), indicating that action mechanism for quinofumelin and quinoxyfen differ.

(7) Line 84:Please introduce why RNA-Seq was designed in the study first. What were the groups compared? How was the experiment set up? Without this background, it is hard to know why and how you did the experiment.

According to your suggestions, we have added the description in Section Results. In addition, the experimental process was described in Section Materials and methods as follows: A total of 20 mL of YEPD medium containing 1 mL of conidia suspension (1×105 conidia/mL) was incubated with shaking (175 rpm/min) at 25°C. After 24 h, the medium was added with quinofumelin at a concentration of 1 μg/mL, while an equal amount of dimethyl sulfoxide was added as the control (CK). The incubation continued for another 48 h, followed by ﬁltration and collection of hyphae. Carry out quantitative expression of genes, and then analyze the differences between groups based on the results of DESeq2 for quantitative expression.

(8) Figures:The figure labeling is missing (Figures 1,2,3 etc). Please re-order your figure to match the text

The figures have been inserted.

(9) Line. 97:"Volcano plot" is a common plot to visualize DEGs, you can directly refer to the name.

We have revised the description.

(10) Figure 1d, 1e:Can you separate down- and up-regulated genes here? Does the count refer to gene number?

The expression information for down- and up-regulated genes is presented in Figure 1a and 1b. However, these bubble plots do not distinguish down- and up-regulated genes. Instead, they only display the significant enrichment of differentially expressed genes in specific metabolic pathways. To more clearly represent the data, we have added the detailed counts of down- and up-regulated genes for each metabolic pathway in Supplementary Table S1 and S2. Here, the term "count" refers to differentially expressed genes that fall within a certain pathway.

(11) Line 111:Again, no reasoning or description of why and how the experiment was done here.

Based on the results of KEGG enrichment analysis, DEMs are associated with pathways such as thiamine metabolism, tryptophan metabolism, nitrogen metabolism, amino acid sugar and nucleotide sugar metabolism, pantothenic acid and CoA biosynthesis, and nucleotide sugar production compounds synthesis. To specifically investigate the metabolic pathways involved action mechanism of quinofumelin, we performed further metabolomic experiments. Therefore, we have added this description according the reviewer’s suggestions.

(12) Figure 2a:It seems many more metabolites were reduced than increased. Is this expected? Due to the antifungal activity of this compound, how sick is the fungus upon treatment? A physiological study on F. graminearum (in a dose-dependent manner) should be done prior to the omics study. Why do you think there's a stark difference between positive and negative modes in terms of number of metabolites down- and up-regulated?

Quinofumelin demonstrates exceptional antifungal activity against *Fusarium graminearum*. The results indicate that the number of reduced metabolites significantly exceeds the number of increased metabolites upon quinofumelin treatment. Mycelial growth is markedly inhibited under quinofumelin exposure. Prior to conducting omics studies, we performed a series of physiological and biochemical experiments (refer to Qian Xiu's dissertation https://paper.njau.edu.cn/openfile?dbid=72&objid=50_49_57_56_49_49&flag=free). Upon quinofumelin treatment, the number of down-regulated metabolites notably surpasses that of up-regulated metabolites compared to the control group. Based on the findings from the down-regulated metabolites, we conducted experiments by exogenously supplementing these metabolites under quinofumelin treatment to investigate whether mycelial growth could be restored. The results revealed that only the exogenous addition of uracil can restore mycelial growth impaired by quinofumelin.

Quinofumelin exhibits an excellent antifungal activity against *F. graminearum*. At a concentration of 1 μg/mL, quinofumelin inhibits mycelial growth by up to 90%. This inhibitory effect indicates that life activities of *F. graminearum* are significantly disrupted by quinofumelin. Consequently, there is a marked difference in down- and up-regulated metabolites between quinofumelin-treated group and untreated control group. The detailed results were presented in Figures 1 and 2.

(13) Figure 2e:This is a good analysis. To help represent the data more clearly, the authors can consider representing the expression using fold change with a p-value for each gene.

To more clearly represent the data, we have incorporated the information on significant differences in metabolites in the de novo pyrimidine biosynthesis pathway, as affected by quinofumelin, in accordance with the reviewer’s suggestions.

(14) Line 142:Please indicate fold change and p-value for statistical significance. Did you validate this by RT-qPCR?

We validated the expression level of the DHODH gene under quinofumelin treatment using RT-qPCR. The results indicated that, upon treatment with the EC50 and EC90 concentrations of quinofumelin, the expression of the DHODH gene was significantly reduced by 11.91% and 33.77%, respectively (P<0.05). The corresponding results have been shown in Figure S4.

(15) Line 145:It looks like uracil is the only metabolite differentially abundant in the samples - how did you conclude this whole pathway was impacted by the treatment?

The experiments involving the exogenous supplementation of uracil revealed that the addition of uracil could restore mycelial growth inhibited by quinofumelin. Consequently, we infer that quinofumelin disrupts the de novo pyrimidine biosynthesis pathway. In addition, as uracil is the end product of the de novo pyrimidine biosynthesis pathway, the disruption of this pathway results in a reduction in uracil levels.

(16) Figure 3:What sequence was used as the root of the tree? Why were the species chosen? Since the BLAST query was *Homo sapiens* sequence, would it be good to use that as the root?

FgDHODHII sequence was used as the root of the tree. These selected fungal species represent significant plant-pathogenic fungi in agriculture production. According to your suggestion, we have removed the BLAST query of *Homo sapiens* in Figure 3.

(17) Figure 4:How were the concentrations used to test chosen?

Prior to this experiment, we carried out concentration-dependent exogenous supplementation experiments. The results indicated that 50 μg/mL of uracil can fully restore mycelial growth inhibited by quinofumelin. Consequently, we chose 50 μg/mL as the testing concentration.

(18) Line 164:Why do you hypothesize supplementing dihydroorotate would restore resistance? The metabolite seemed accumulated in the treatment condition, whereas downstream metabolites were comparable or even depleted. The DHODH gene expression was suppressed. Would accumulation of dihydroorotate be associated with growth inhibition by quinofumelin? Please include the hypothesis and rationale for the experimental setup.

DHODH regulates the conversion of dihydroorotate to orotate in the de novo pyrimidine biosynthesis pathway. The inhibition of DHODH by quinofumelin results in the accumulation of dihydroorotate and the depletion of the downstream metabolites, including UMP, uridine and uracil. Consequently, downstream metabolites were considered as positive controls, while upstream metabolite dihydroorotate served as a negative control. This design further demonstrates DHODH as action target of quinofumelin against *F. graminearum*. In addition, the accumulation of dihydroorotate is not associated with growth inhibition by quinofumelin; however, but the depletion of downstream metabolites in the de novo pyrimidine biosynthesis pathway is closely associated with growth inhibition by quinofumelin.

(19) Line 168:I'm not sure if this conclusion is valid from your results in Figure 4 showing which metabolites restore growth.

o minimize the potential influence of strain-specific effects, five strains were tested in the experiments shown in Figure 4. For each strain, the first row (first column) corresponds to control condition, while second row (first column) represents treatment with 1 μg/mL of quinofumelin, which completely inhibits mycelial growth. The second row (second column) for each strain represents the supplementation with 50 μg/mL of dihydroorotate fails to restore mycelial growth inhibited by quinofumelin. In contrast, the second row (third column, fourth column, fifth colomns) for each strain demonstrated that the supplementation of 50 μg/mL of UMP, uridine and uracil, respectively, can effectively restore mycelial growth inhibited by quinofumelin.

(20) Figure 5a:The fact you saw growth of the deletion mutant means it's not lethal. However, the growth was severely inhibited.

Our experimental results indicate that the growth of the deletion mutant is lethal. The mycelial growth observed originates from mycelial plugs that were not exposed to quinofumelin, rather than from the plates amended with quinofumelin.

(21) Figure 5b:Would you expect different restoration of growth in the presence of quinofumelin vs. no treatment? The wild type control is missing here. Any conclusions about the relationship between quinofumelin, FgDHODHII, and other metabolites in the pathway?

Under no treatment with quinofumelin, mycelial growth remains normal and does not require restoration. In the presence of quinofumelin treatment, the supplementation of downstream metabolites in the de novo pyrimidine biosynthesis pathway can restore mycelial growth that is inhibited by quinofumelin. The wild-type control group is illustrated in Figure 4. Figure 5b depicts the phenotypes of the deletion mutants. With respect to the relationship among quinofumelin, FgDHODHII, and other metabolites, quinofumelin specifically targets the key enzyme FgDHODHII in the de novo pyrimidine biosynthesis pathway, disrupting the conversion of dihydroorotate to orotate, which consequently inhibits the synthesis downstream metabolites including uracil.

(22) Figure 6b:Lacking positive and negative controls (known binder and non-binder). What does the Kd (in comparison to other interactions) indicate in terms of binding strength?

We tested the antifungal activities of publicly reported DHODH inhibitors (such as leflunomide and teriflunomide) against *F. graminearum*. The results showed that these inhibitors exhibited no significant inhibitory effects against the strain PH-1. Therefore, we lacked an effective chemical for use as a positive control in subsequent experiments. Biacore experiments offers detailed insights into molecular interactions between quinofumelin and DHODHII. As shown in Figure 6b, the left panel illustrates the time-dependent kinetic curve of quinofumelin binding to DHODHII. Within the first 60 s after quinofumelin was introduced onto the DHODHII surface, it bound to the immobilized DHODHII on the chip surface, with the response value increasing proportionally to the quinofumelin concentration. Following cessation of the injection at 60 s, quinofumelin spontaneously dissociated from the DHODHII surface, leading to a corresponding decrease in the response value. The data fitting curve presented on the right panel indicates that the affinity constant KD of quinofumelin for DHODHII is 6.606×10-6 M, which falls within the typical range of KD values (10-3 ~ 10-6 M) for protein-small molecule interaction patterns. A lower KD value indicates a stronger affinity; thus, quinofumelin exhibits strong binding affinity towards DHODHII.

**Reviewer #3 (Recommendations for the authors):**
The authors should add information about the other molecule within this class, ipflufenoquin, and what is known about it. There are already published data on its mode of action on DHODH and the role of pyrimidine biosynthesis.

We have added the information regarding action mechanism of ipflufenoquin against filamentous fungi in discussion section.

The work of Higashimura et al is not appreciated well enough as they already showed the role of quinofumelin upon DHODH II.

We sincerely appreciate the reviewer's insightful comment regarding the work of Higashimura et al. We agree that their investigation into the role of quinofumelin in DHODH II inhibition provides critical foundational insights for this field. In the revised manuscript, we have incorporated the reference in the introduction section and expanded the discussion of their work in the discussion section to more effectively contextualize their contributions.

It is unclear how the protein model was established and this should be included. What species is the molecule from and how was it obtained? How are they different from Fusarium?

The three-dimensional structural model of *F. graminearum* DHODHII protein, as predicted by AlphaFold, was obtained from the UniProt database. Additionally, a detailed description along with appropriate citations has been incorporated in the ‘Manuscript’ file.